



# Optimizing a twin-chamber system for direct ozone production rate
measurement
Yaru Wang[1#], Yi Chen[2#], Suzhen Chi[1#], Jianshu Wang[1], Chong Zhang[1], Weixiong Zhao[3], Weili Lin[2],
Chunxiang Ye[1*]
[1]State Key Joint Laboratory for Environmental Simulation and Pollution Control, Center for Environment and Health, and
College of Environmental Sciences and Engineering, Peking University, Beijing, 100871, China.
[2]Key Laboratory of Ecology and Environment in Minority Areas (Minzu University of China), National Ethnic Affairs
Commission, Beijing, 100081, China.
[3]Laboratory of Atmospheric Physico-Chemistry, Chinese Academy of Sciences Hefei Institutes of Physical Science Anhui
Institute of Optics and Fine Mechanics, Chinese Academy of Sciences, Hefei, 230031, Anhui, China.
[#]Yaru Wang, Yi Chen and Suzhen Chi contribute equally to this work.
*Correspondence to*: Chunxiang Ye (c.ye@pku.edu.cn)
A manuscript submitted to *AMT*


**Abstract.** High Ozone Production Rate (OPR) leads to $O_3$ pollution episodes and adverse human health outcomes.
Discrepancies between OPR observation (Obs-OPR) and OPR modeling (Mod-OPR) as calculated from observed and modeled
peroxy radical and nitrogen oxides reminds of a yet-perfect understanding of $O_3$ photochemistry. Direct measurement of OPR
(Mea-OPR) by a twin-chamber system emerges with the optimization required for suppressing the wall effect. Herein, we
minimized the chamber surface area to volume ratio (S/V) to 9.8 $m^{-1}$ and the dark uptake coefficient of $O_3$ to the order of $10^{-9}$.
Condition experiments further revealed a photo-enhanced $O_3$ uptake and recommended an essential correction. We finally
characterized a measurement uncertainty of ±27% and a detection limit of 2.8 ppbv $h^{-1}$ (3SD), which suggests that Mea-OPR
is sensitive enough to measure OPR in urban or suburban environments. Application of this system in urban Beijing during
the Winter Olympic Games recorded a noontime OPR of 7.4 (±3.8, 1SD) ppbv $h^{-1}$, which indicates fairly active $O_3$
photochemistry despite the pollution control policy implemented. Mea-OPR *versus* $j(O^1D)$ slope of $6.1 \times 10^5$ ppbv $h^{-1}$ $s^{-1}$
confirmed fairly active $O_3$ photochemistry, which was assisted by a high abundance of VOCs and $NO_x$, atypically high Mea-
OPR even under high-$NO_x$ conditions, but mediated by relatively weak ultraviolet (UV) radiation.

**Short summary.** We reported an optimized system (Mea-OPR) for direct measurement of ozone production rate, which
showed a precise, sensitive and reliable measurement of OPR for at least urban and suburban atmosphere, and active $O_3$
photochemical production in winter Beijing. Herein, the Mea-OPR system also shows its potential in exploring the fundamental
$O_3$ photochemistry, i.e., surprisingly high ozone production even under high-$NO_x$ conditions.















## 1 Introduction

Tropospheric ozone ($O_3$) is a hazardous air pollutant and a key product of photochemical smog (Prinn, 2003; Sillman, 2003).
The growing abundance of $O_3$ due to primarily photochemical production has long been associated with the human health risk
(Ho et al., 2007), plant growth issues (Ashmore, 2005), and climate change (Forster et al., 2007). Understanding why the $O_3$
level continues increasing in some regions, despite rigorous emissions control policy (Tarasick et al., 2019), is vital to reverse
the $O_3$ increasing trend.
In favorable meteorological scenarios for high ozone production rate such as no or light wind under strong solar radiation,
$O_3$ pollution episodes have been repeatedly recorded. The mixing ratio of $O_3$ (8-hour maximum average) could reach over 100
ppbv, doubling or tripling its background values. Photochemical production reactions of $O_3$ mainly involve volatile organic
compounds (VOCs), nitrogen oxides ($NO_x = NO+NO_2$) and solar radiation (Finlayson-Pitts and Jr. Pittes, 1999; Seinfeld and
Pandis, 2006). $NO_2$ photolysis produces NO and $O_3$ while $O_3$ reacts with NO to recycle $NO_2$. These rapidly cycling processes
(within minutes) are defined as the Leighton cycle. Under the photostationary-state (PSS) assumption for the Leighton cycle,
neither $O_3$ production nor loss is occurring. Production rate of $O_3$, referred to as $P(O_3)$, increases as peroxy radicals ($HO_2+RO_2$)
efficiently oxidize NO to produce extra $NO_2$ (Eq. 1). Additionally, photochemical consumption of $O_3$, referred to as $L(O_3)$ (Eq.
2), represents a considerable fraction of a comprehensive chemical budget of $O_3$ to offset $O_3$ production:
$$P(O_3) = k_{HO_2+NO}[HO_2][NO] + \sum_i (\alpha_i k_{RO_{2i}+NO}[RO_{2i}][NO]) \tag{1}$$
$$L(O_3) = f_{O(^1D)+H_2O} j_{O_3}[O_3] + k_{OH+O_3}[OH][O_3] + k_{HO_2+O_3}[HO_2][O_3] + \sum_i (k_{O_3+Alkene_i}[O_3][Alkene_i]) +$$
$$k_{OH+NO_2}[OH][NO_2] + L(O_3+halogens) \tag{2}$$
$$OPR = P(O_3) - L(O_3) \tag{3}$$
where $k_{RO_{2i}+NO}$ defines the rate coefficient of the reaction between a specific peroxy radical and NO. Alkyl nitrate formation
from molecule isomerization of [R-O-O-NO] competes with $O_3$ production. The yield of $O_3$ production is defined as $\alpha_i$ for a
specific $RO_2$ as it reacts with NO, while corresponding yield of alkyl nitrate is defined as $1 - \alpha_i$. $f_{O(^1D)+H_2O}$ is defined as the
probability of an oxygen atom, a photo-fragment of $O_3$ photolysis, reacting with water vapor to produce two OH radicals. OPR
represents the effective or net production rate of $O_3$.
$NO_x$ and VOCs regulate the photochemical cycling of radicals and therefore OPR. Under low-$NO_x$ conditions, bimolecular
reactions among peroxy radicals buffer the accumulation and chain propagation of peroxy radicals. Increasing abundance of
$NO_x$ thus favors chain propagation of peroxy radicals, outcompetes bimolecular reactions among peroxy radicals and favors
photochemical production of $O_3$. Under high-$NO_x$ conditions, chain termination reaction of $NO_2$ with OH radical, one major
route of $O_3$ photochemical consumption, competes with the chain propagation of peroxy radicals and photochemical production
of $O_3$. This scenario analysis reveals the nonlinear relationship between OPR and its precursor of $NO_x$ (Cazorla et al., 2012;
Guo et al., 2021). The role of VOCs in $O_3$ production and the common sources of both VOCs and $NO_x$ in some areas further



complicate the OPR-$NO_x$ relationship and its temporal-spatial variability (Schroeder et al., 2017). High time resolution
characterization of OPR in the specific atmosphere of concern will be essentially helpful to characterize OPR-$NO_x$ relationship
and its environmental variability.
In view of aforementioned expressions, Obs-OPR or Mod-OPR can be obtained from the measured or model-simulated $HO_2$
and $RO_2$, given mixing ratio of NO is usually measured in field campaigns (Ren et al., 2003, 2013; Green et al., 2006; Kanaya,
2002; Griffith et al., 2015; Tan et al., 2019b). A $NO_x$-dependent representation of Mod-OPR was now summarized with
consistent and considerable underestimation in high-$NO_x$ areas (Ren et al., 2013; Tan et al., 2019b, 2017; Whalley et al., 2018,
2021; Brune et al., 2016) while slightly underestimation or even possible overestimation in low-$NO_x$ areas (Ren et al., 2003;
Whalley et al., 2018). For instance, Ren et al. developed an observation-prescribed box model and outputted a underestimated
Mod-OPR by up to one order of magnitude compared with Obs-OPR when the median NO mixing ratio reached 17 ppbv (Ren
et al., 2013), while overestimated Mod-OPR by 1.4–1.7 times when NO was below 1.0 ppbv was also reported (Ren et al.,
2003). Evidently, such observation-model discrepancy on OPR decreased our confidence for understanding both the $O_3$ budget
and policy efficiency of regional $O_3$ pollution control.
The idea of direct measurement of OPR, Mea-OPR, which dated as far back as 1971, was first proposed by Jeffries (1971).
The principle is based on differential $O_3$ signal when ambient air continuously flows into two identical and horizontally
oriented chambers, one of which (called reaction chamber) is enough transparent for UV and visible radiation to mimic ambient
photochemistry and the other one (named reference chamber) is covered or coated by UV-blocking film to filter UV to suppress
the $O_3$ photochemical production (Cazorla and Brune, 2010; Baier et al., 2015; Sadanaga et al., 2017; Sklaveniti et al., 2018).
As simultaneous measurements of $NO_2$ and $O_3$ (= $O_x$) in the reaction chamber and reference chamber can cancel out any ozone
difference resulting from the rapid interconversion between $NO_2$ and $O_3$ in two chambers, a differential $O_x$ signal rather than
$O_3$ was further adopted. We refer to the twin-chamber method for direct measurement of OPR or the measurement result by
such method as Mea-OPR in the context.
Accurate and precise Mea-OPR relies on several assumptions implied from the twin-chamber measurement scheme. The
most important one is that photochemistry in the reaction chamber should be able to mimic the ambient condition while $O_3$
photochemistry in the reference chamber is totally suppressed. Several versions of Mea-OPR, or named MOPS in literature,
have been managed to mimic photochemistry in both chambers. Teflon film and quartz glass are usually adopted for the
chamber body to guarantee UV transmittance in the reaction chamber (Cazorla and Brune, 2010; Sadanaga et al., 2017;
Sklaveniti et al., 2018). Light transmittance achieves easily above 90% for either Teflon (FEP Teflon film, 0.05 mm thick) or
quartz tube, providing ideal UV conditions in the reaction chamber. UV blocking film is also effective in creating a "dark"
condition for $O_3$ photochemistry.
Another precondition for Mea-OPR is that measurement of $O_x$ could precisely represent the differential $O_x$ signal, $\Delta O_x$,
between chambers. Measurement uncertainties in $O_x$ might be enlarged as transferred to be the measurement uncertainties in
$\Delta O_x$, thus further amplifying the measurement uncertainties of Mea-OPR in a unit of ppbv $h^{-1}$ because the gas residence time
in Mea-OPR chambers is typically much shorter than 1 hour. Both insensitivity and the drift in measurement baseline of $O_3$





and NO₂ instruments contribute to the measurement uncertainties of $\Delta O_x$ (Cazorla and Brune, 2010; Baier et al., 2015). Such
stringent requirement in precise $\Delta O_x$ measurements requires advanced and tested measurement techniques and well trained
technicians to perform the measurement, which might limit the widespread deployment of Mea-OPR. Reliable and sensitive
measurement technique for $O_3$ and $NO_2$ might be a savior in this situation.
Last but not least, the chamber wall effect must be suppressed and quantified. The chamber wall effect might change the
abundance of $O_x$, which presents false signal of $\Delta O_x$ that is difficult to be decoupled from photochemical $\Delta O_x$. The chamber
wall effect might also change the abundance of photochemical intermediates, leading to unwanted perturbation on $O_3$
photochemistry in the reaction chamber. Sklaveniti et al. (2018) estimated that $O_3$ uptake loss in their version of Mea-OPR
would lead to false Mea-OPR signal of ~20 ppbv h⁻¹ assuming ambient $O_3$ to be 50 ppbv. It was also found that HONO
production in uptake experiment of $NO_2$ on quartz chamber and Teflon chamber can reach up to tens of ppbv h⁻¹ under
irradiated conditions, which would lead to an overestimation of Mea-OPR by approximately 27% on average (Sklaveniti et
al., 2018) or Mea-OPR error of around 10 ppbv h⁻¹ (Baier et al., 2015). Effective design to suppress the wall effect is, therefore,
the key to ensure the data quality of Mea-OPR. Previous designs are devoted to keeping the air in plug flow motion and a
small-diameter (14.0–17.8 cm) chamber is usually chosen for it is easier to manipulate (Cazorla and Brune, 2010; Baier et al.,
2015; Sadanaga et al., 2017; Sklaveniti et al., 2018). However, a small diameter works against the wall effect suppression. The
S/V ratio was decreased by 20% by employing a larger chamber volume (26.9 L) in the second version of Mea-OPR (Baier et
al., 2015), relative to the first generation (11.3 L) (Cazorla and Brune, 2010). Referring to the uptake coefficient calculation
formula (Eq. 4), a 20% decrease of S/V will result in a 20% reduction in false Mea-OPR signal due to uptake loss of $O_3$ for a
given uptake coefficient and $O_3$ abundance. Previous research has also confirmed less $O_x$ loss on Teflon wall surface (Sadanaga
et al., 2017) than on quartz surface (Sklaveniti et al., 2018). Sadanaga et al. (2017) first managed to coat the inner wall surface
of the quartz chamber with transparent Teflon and effectively reduced the wall effect.
$\gamma_{O_3} = \dfrac{4 \times \Delta O_{3,\, uptake}}{O_{3,\, amb} \times \omega_{O3} \times \tau \times S/V}$                    (4)
To date, these effective designs have not yet been integrated and evaluated in a state-of-the-art version of Mea-OPR. In this
article, we will present our construction of a state-of-the-art Mea-OPR and condition experiments to characterize this Mea-
OPR. Later, the employment of Mea-OPR system in Beijing, a megacity in China, provides further validation of it.
**2 Experimental section**
**2.1 Construction of state-of-the-art Mea-OPR**
The schematic of our Mea-OPR is shown in Fig. S1. The quartz chamber inner wall was coated with the transparent Teflon.
It was chosen over the Teflon film chamber for Teflon film chamber showed its weakness in working in high wind velocity
conditions. The quartz chamber and the Teflon coating together presented an ideal transmittance (> 88%) for both UV and





visible light (Fig. S2). The difference in $j(O^1D)$ between the ambient and the reaction chamber calculated by measured solar
flux (Metcon CCD-Spectrograph) in Beijing winter was within 4% (Table S1). UV filter was adopted for the reference chamber
(transmittance for UV was 0, Fig. S3) and enabled small temperature and RH differences in both chambers (Fig. S4).
Parallel and accurate measurements of $\Delta O_x$ were supported by two identical sets of $O_3$ analyzer (Thermo Scientific, Model
49i, LOD: 1.0 ppbv) and $NO_2$ analyzer (Los Gatos Research, Inc., Model 911-0009, LOD: 50 pptv). To calculate the wall loss
of $O_x$ in both chambers in real time, another set of $O_3$ analyzer (Thermo Scientific, Model 49i, LOD: 1.0 ppbv) and homemade
iBBCES-$NO_2$ instrument (AIOFM, LOD: 168 pptv; 30 min background shifted: 100 pptv) (Fang et al., 2017) were sampling
the ambient air. The system was running with a duty cycle in a 23 h plus 1 h, which included 1 h instrument alignment as all
instruments shifted to the ambient sampling line at midnight. The purpose of the instrument alignment was to check the
working status of the measurements on a daily base, rather than data correction. $\Delta O_x$ up to 1 ppbv in a 1 h alignment reminded
of instrument calibration and maintenance, which were otherwise conducted on a weekly basis. Under the working protocol,
the system performed stably in a 1-month measurement duty from February 5 to March 12, 2022. Multiple calibrations
suggested the instrument response shifted within ±0.4% and ±1.9% for $O_3$ and $NO_2$, respectively, in this period (Table S2).
The instrument alignment experiment suggested that nigttime $\Delta O_x$ was 0.07 (±0.26) ppbv, within the instrument detection
limits (Fig. S5).
Our Mea-OPR development put emphasis on wall effect management. The diameter of our Mea-OPR was 41 cm, which
produced a S/V ratio that was 1.8 fold less than the lowest value ever published in the literature (Cazorla and Brune, 2010;
Baier et al., 2015). This reduced the $O_3$ wall effect by 1.8 folds even though the same uptake coefficient of $O_3$ was taken into
consideration (Eq. 4), and so did for wall effect of other species, such as $NO_2$ and HONO. Larger diameter of our Mea-OPR
also gave the flexibility of the working flow rate, which spanned from 5 to 20 L min$^{-1}$ while enabling multiple sampling
instruments equipped and a residence time of up to 30 min in a plug flow mode (Fig. S1). Another key design to suppress the
wall effect was the transparent Teflon coating, which was essential for accurate Mea-OPR by reducing the uptake coefficient
of $O_3$ from $10^{-8}$ on quartz wall (Sklaveniti et al., 2018) to $10^{-9}$ on Teflon coating wall under dark conditions.
Mea-OPR can be calculated in Eq. (5), wherein $\Delta NO_2$ and $\Delta O_3$ are the difference values of $NO_2$ and $O_3$ between both
chambers, respectively. $\tau$ is the mean gas residence time in chambers (Fig. S6). $\gamma$ is the uptake coefficient of $O_3$ in the
chamber. $\omega$ represents the mean molecular velocity of $O_3$ in m s$^{-1}$. D is the diameter of chambers in m. $O_{3,\,amb}$ represents
the ambient $O_3$ concentration in ppbv. $\varphi_{trans}$ is the ratio of in-chamber $j(O^1D)$ to ambient $j(O^1D)$ as determined by the UV
transmittance of Mea-OPR system. $\varphi_{\Delta HONO\ or\ \Delta NOx}$ is the ratio of OPR in the reaction chamber to that in the ambient owing to
the presence of $\Delta HONO$ or $\Delta NO_x$ between the reaction chamber and the ambient.
$$\text{Mea-OPR} = \left( \frac{\Delta NO_2 + \Delta O_3}{\tau} + \frac{(\gamma_{Rea} \cdot \omega_{Rea} - \gamma_{Ref} \cdot \omega_{Ref}) \cdot O_{3,\,amb}}{D} \right) \cdot \frac{1}{\varphi_{trans} \cdot \varphi_{\Delta HONO\ or\ \Delta NOx}} \qquad (5)$$



**2.2 Condition experiments to characterize wall effect**

As Mea-OPR was sampling the ambient air, $O_3$ wall loss and photochemical production were occurring simultaneously and, thus, cannot be decoupled from each other. A condition experiment of zero-$NO_x$-and-zero-VOCs (referred to as zero-$NO_x$-and-high-$O_3$ experiment) was designed and, thus, zero OPR was assumed. High $O_3$ control (also referred to as ambient $O_3$ for simplicity) in zero-$NO_x$-and-high-$O_3$ experiment favored measurement of $O_3$ wall losses directly in both chambers on 6 March, 2022. In addition, HONO production from $NO_2$ uptake (Eq. 6) on the Teflon film and quartz surface have been proposed by Baier et al. (2015) and Sklaveniti et al. (2018) as a potential perturbation on photochemical production of $O_3$ in the reaction chamber. To obtain differential $NO_x$ and HONO signals in chambers relative to the ambient, additional measurements of HONO in the chambers and in the ambient were equipped in a 1-week HONO production experiment as Mea-OPR was deployed in 11–18 February, 2022. The above condition experiments to characterize wall effect were described in detail in the Supplement (S2.6).

$$2NO_2 + H_2O \text{ (ads)} \rightarrow HONO + HNO_3 \qquad\qquad (6)$$

**2.3 Field application of Mea-OPR**

Our Mea-OPR system was deployed on the top floor of an academic building on the campus of Peking University (39°59′23″ N, 116°18′25″ E). The observatory as a typical urban and polluted site in Beijing City, China, was impacted by considerable fresh, anthropogenic emissions in the surroundings, such as the 4[th] ring-road traffic emission. More details about this site were also described elsewhere (Guo et al., 2010; Tang et al., 2018). Notably, the measurement period coincided with the Beijing 2022 Winter Olympic Games from 4 to 20 February, when the Municipal Government of Beijing implemented a package of factory and transportation controls to improve air quality. Owing to the aggressive control measures and favorable meteorological conditions, no haze events and substantial reductions in gas pollutant concentrations were observed (Guo et al., 2023; Liu et al., 2022). Therefore, the air pollution conditions during this period can be regarded as relatively clean compared to those polluted episodes in the same period of other years.

**3 Results and discussion**

**3.1 $O_3$ uptake evaluation**

A zero-$NO_x$-and-high-$O_3$ experiment was conducted for directly measuring the $O_3$ wall loss in both chambers with little perturbation from $O_3$ photochemical production in chambers. As shown in Fig. 1a, evident $O_3$ wall loss ($\Delta O_{3, raw} = O_3$ in the ambient $- O_3$ in the chamber) were observed and higher $O_3$ wall loss was observed for the reaction chamber ($\Delta O_{3, raw} = 7.7$ ppbv) relative to the reference chamber ($\Delta O_{3, raw} = 2.3$ ppbv) at noon. The ambient $O_3$ control was measured at approximately 113.0 ppbv just before entering chambers. $NO_x$ concentration was measured around 0.03–1.05 ppbv to double-check the low level of $NO_x$ in the control experiment. A slight increase in $NO_x$ from the morning to the noon was accompanied with increasing





$j(O^1D)$, which might be attributed to the previously-identified unknown source of HONO and $NO_x$ in Teflon chamber (Zhou
et al., 2003). The unknown chamber source of HONO was also confirmed later during the 1-week HONO production
experiment (see below). The MCM model was conducted to calculate $O_3$ production in chambers. $NO_x$ and CO were both
major prescribed chemical parameters. At noon, the corresponding $O_3$ production in the reaction chamber, $\Delta O_{3,\ photochemistry}$,
was found to be 2.7 ppbv in the residence time of 20 min (Fig. S7), which comprises 35% of the $\Delta O_{3,\ raw}$ for the reaction
chamber. $O_3$ production was negligible in the reference chamber at noon because stray light in the reference chamber was too
weak to be meaningful for $O_3$ photochemistry.
After eliminating the photochemical contribution, $\Delta O_{3,\ uptake}$ (= $\Delta O_{3,\ raw} - \Delta O_{3,\ photochemistry}$) was obtained and the uptake
coefficient of $O_3$ on chambers could be calculated according to Eq. (4). The uptake coefficient from a typical value of $7.11 \times$
$10^{-8}$ on quartz wall (Sklaveniti et al., 2018) (Table S3) was successfully reduced to the present average value of $8.12 \times 10^{-9}$
on the transparent Teflon coating under dark conditions. Under typical working conditions of Mea-OPR, $O_3$ uptake loss
contributes to a false Mea-OPR signal of 20.3 ppbv h$^{-1}$ at uptake coefficient of $7.11 \times 10^{-8}$ and S/V ratio of 18 m$^{-1}$ (the least
in the literature), relative to a false Mea-OPR signal of 1.29 ppbv h$^{-1}$ at uptake coefficient of $8.12 \times 10^{-9}$ and S/V ratio of 9.76
m$^{-1}$ in our Mea-OPR, assuming ambient $O_3$ concentration of 50 ppbv. This calculation suggests the essential success in wall
effect suppression by our strategy, i.e., to minimize chamber S/V ratio and uptake coefficient of $O_3$ on the chamber wall, for
assuring the data quality of Mea-OPR.
During the daytime, UV-dependent $\gamma$ was observed and inferred a photo-enhanced uptake of $O_3$ of up to $8.98 \times 10^{-8}$ at noon,
nearly one order of magnitude higher relative to $8.12 \times 10^{-9}$ at night (Fig. 1b). Low uptake coefficient of $5.17 \times 10^{-9}$ under
dark condition was observed for Teflon surface due to its inert nature (Sadanaga et al., 2017). However, photosensitization
reactions of $O_3$ on organic coating was observed with an uptake coefficient in the range of $10^{-6}$–$10^{-5}$ under near-UV and visible
irradiation relative to $10^{-7}$–$10^{-6}$ in the dark (Styler et al., 2009; D'Anna et al., 2009). This suggested that the aerosol particles
deposited on the inner wall surface might substantially contribute to the photo-enhanced wall loss of $O_3$. A routine water flush
cleaning and UV-photochemical-aging cleaning of both chambers were then scheduled after occurrence of heavy pollution
episodes ($PM_{2.5} > 80\ \mu g\ cm^{-3}$) (Juda-Rezler et al., 2020). RH might modify aerosol phase state and thus was another parameter
affecting the uptake coefficient on aerosol particles and apparent wall loss of $O_3$. The daytime RH was typically below 50%,
approximately the threshold RH of particle phase shift in the city of Beijing (Liu et al., 2016). A relatively stable $\gamma$ at low RH
was also observed. The daytime $\gamma$ was then expressed as a function of merely $j(O^1D)$, not considering the complex influence
from RH variation below 50% (Fig. 1c). An increasing $\gamma$ as a function of RH for the nighttime was summarized, with a
threshold RH of *ca.* 60% (Fig. 1d).



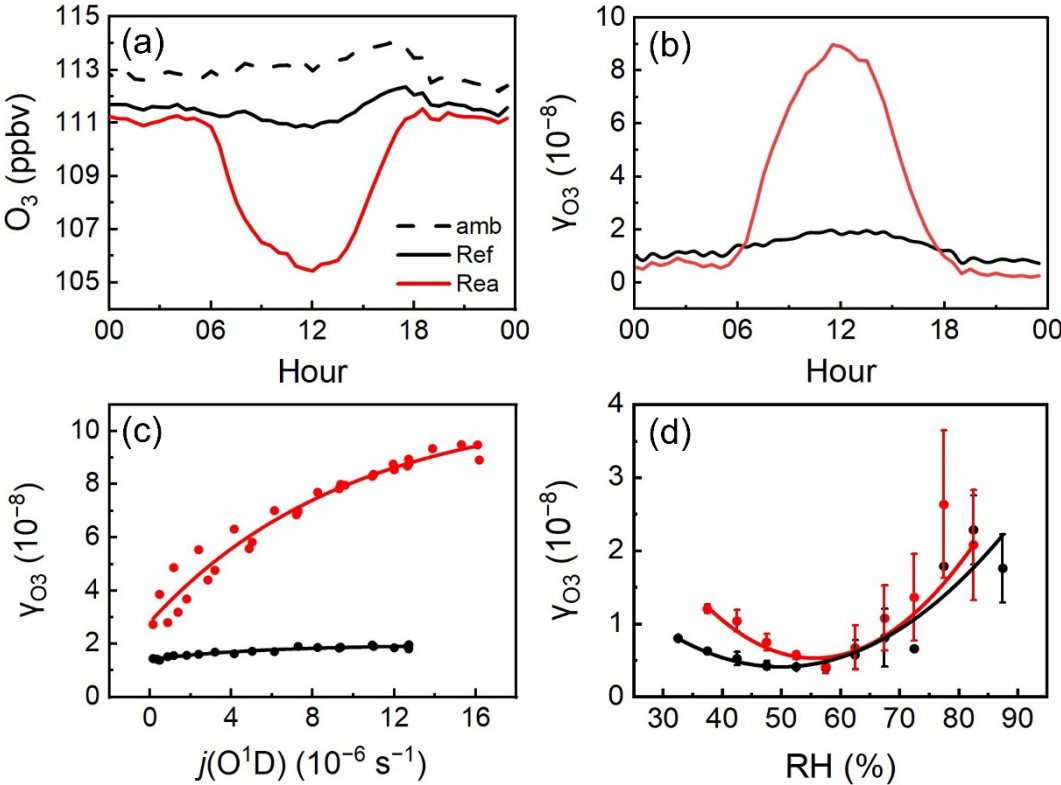

Figure 1: Results of condition experiment designed to measure $O_3$ wall loss. (a) $O_3$ concentrations in the ambient (approximately 113.0 ppbv) and in both chambers as zero-$NO_x$-and-high-$O_3$ air was introduced to both chambers; (b) Calculated half-hour resolution uptake coefficient ($\gamma$) of $O_3$ in both chambers; (c) Fitted exponential relationship between $\gamma$ and $j(O^1D)$ in daytime, i.e., $j(O^1D) > 1 \times 10^{-7}$ s$^{-1}$ ($R^2 = 0.93$ and $0.96$, respectively); (d) Fitted polynomial relationship between $\gamma$ and RH during night, i.e., $j(O^1D) < 1 \times 10^{-7}$ s$^{-1}$ ($R^2 = 0.82$ and $0.82$, respectively). Error bar represents standard deviation of $\gamma$ in each RH interval of 5%. Fitting method is ordinary least squares estimation.

Sklaveniti et al. (2018) also suspected photo-enhanced $O_3$ uptake in the reaction chamber as a major measurement interference for their Mea-OPR system utilizing quartz wall without Teflon coating. However, no quantification and correction of Mea-OPR interference from the $O_3$ uptake was suggested. The $j(O^1D)$-dependent $\gamma$ for the daytime and RH-dependent $\gamma$ for the nighttime as determined in our control experiments provided the first reliable correction for Mea-OPR (Eqs. 7–10). For the correction, $O_3$ concentrations measured in the ambient could be used to calculate the $\Delta O_{3, uptake}$ (Eq. 4). After multiple control experiments (not shown), we could also assume the uptake coefficient of $O_3$ being stable between two adjacent control experiments.

$$\gamma_{Ref} = -5.53 \times 10^{-9} \times \exp\left(-\frac{j(O^1D)}{5.41 \times 10^{-6}}\right) + 1.96 \times 10^{-8} \qquad (7)$$

$$\gamma_{Rea} = -8.21 \times 10^{-8} \times \exp\left(-\frac{j(O^1D)}{9.76 \times 10^{-6}}\right) + 1.10 \times 10^{-7} \qquad (8)$$

$$\gamma_{Ref} = 1.28 \times 10^{-11} \times RH^2 - 1.28 \times 10^{-9} \times RH + 3.60 \times 10^{-8} \qquad (9)$$





$\gamma_{Rea} = 2.37 \times 10^{-11} \times RH^2 - 2.56 \times 10^{-9} \times RH + 7.50 \times 10^{-8}$              (10)
**3.2 In-chamber HONO production evaluation**
In the 1-week HONO production experiment, $\Delta NO_x$ (= $NO_x$ in the ambient − $NO_x$ in the chamber) showed no obvious
diurnal variation, with an average of 1.01 and 0.77 ppbv in the reaction and reference chamber, respectively (Fig. 2a). Based
on Eq. (S3), the average uptake coefficient of $NO_2$ in the reaction and reference chamber were calculated as $8.66 \times 10^{-8}$ and
$6.43 \times 10^{-8}$, respectively. Despite of considerable uptake coefficient of $NO_2$, compared with that of $O_3$, much lower $NO_2$ level
(12.9 ppbv) relative to $O_3$ inferred negligible wall loss of $NO_2$, which was transferred to *ca.* 1.79 ppbv h$^{-1}$ false signal of Mea-
OPR at most during the 1-week HONO production experiment. The perturbation on $O_3$ photochemistry in the reaction chamber
by such $\Delta NO_x$ was evaluated by MCM model to be negligible due to the insensitive response of $O_3$ photochemistry to $NO_x$
abundance in the range around 10 ppbv.
Uptake loss during the nighttime and production of HONO during the daytime in both chambers were spotted (Fig. 2b).
Nighttime loss of HONO on the wall surface, as observed in the lower HONO in both chambers relative to the ambient HONO
($\Delta HONO = -0.22$ ppbv on average for the reaction chamber), was not suspected even assuming that uptake loss of $NO_2$ and
heterogenous production of HONO were negligible (Sadanaga et al., 2017). However, Sklaveniti et al. (2018) reported HONO
production rate of up to 9 ppbv h$^{-1}$ in controlled ambient concentration of $NO_2$ under dark condition. HONO uptake on
deliquescent aerosol particles at night might account for the HONO loss here (Ren et al., 2020). RH in the reaction chamber
scattered at approximately 61% (±14%) and was much higher than the ambient air of 36% (±14%) during the nighttime, which
might lead to deliquescence of deposited aerosol particles on the wall surface. As the temperature rose and RH dropped in the
early morning, uptake loss of HONO on the wall surface was diminishing. Further decrease in the zenith angle even led to
production or release of HONO in both chambers, resulting in a higher HONO concentration in both chambers relative to the
ambient air during the daytime ($\Delta HONO = 0.09$ ppbv on average for the reaction chamber). Either heterogeneous uptake of
$NO_2$ or unknown temperature-related or UV-related chamber source of HONO or releasing of nighttime uptaken HONO might
account for the daytime $\Delta HONO$ in chambers. Daytime $\Delta HONO$ herein appeared to be less than previous reports in laboratory
condition (Sklaveniti et al., 2018) and in the ambient of Houston (Baier et al., 2015). Both the inert surface coating and less
abundant $NO_2$ in 1-week HONO production experiment might justify the result.
While the small $\Delta NO_2$ and $\Delta HONO$ contribute slightly to the measurement uncertainty of $\Delta O_x$, daytime HONO production
in the reaction chamber might perturb the $O_3$ photochemistry therein and challenge the mimic of $O_3$ photochemistry by the
reaction chamber (Baier et al., 2015; Sklaveniti et al., 2018). We then evaluated the $O_3$ production difference in the reaction
chamber relative to the ambient air. Since the radical budget is closely related to $O_3$ chemistry (Tan et al., 2018a; Ma et al.,
2022), the primary $RO_x$ source budget and $O_3$ production in the reaction chamber and the ambient air were calculated and
compared, based on the MCM models prescribed with HONO in the ambient air and the reaction chamber, respectively. When
the HONO difference between the reaction chamber and the ambient air was maximum in the daytime, HONO photolysis
comprised 22.2% of the total primary $RO_x$ source budget for the ambient air, while constituted 28.2% of the total primary $RO_x$





source budget for the reaction chamber. The overall primary $RO_x$ source budget was at most 8.1% higher in the reaction
chamber than in the ambient air, which revealed that ΔHONO considerably perturbed the $RO_x$ source budget owing to the
considerable contribution from HONO photolysis. Furthermore, $O_3$ production enhancement owing to HONO production in
the reaction chamber, relative to the ambient air, was calculated as 4.5% at most during the 1-week HONO production
experiment (Fig. 2c). It could be seen that although the HONO production in the reaction chamber significantly increased the
$RO_x$ source budget and promoted the $O_3$ production, it might also increase the sink of $O_3$ accordingly, and in general, it had
little effect on the net $O_3$ production. While in less polluted environments, the perturbation of HONO on $O_3$ photochemistry
was assumed to be less and, therefore, negligible. Currently, $NO_2$ uptake and HONO production correction are not applied for
our Mea-OPR.

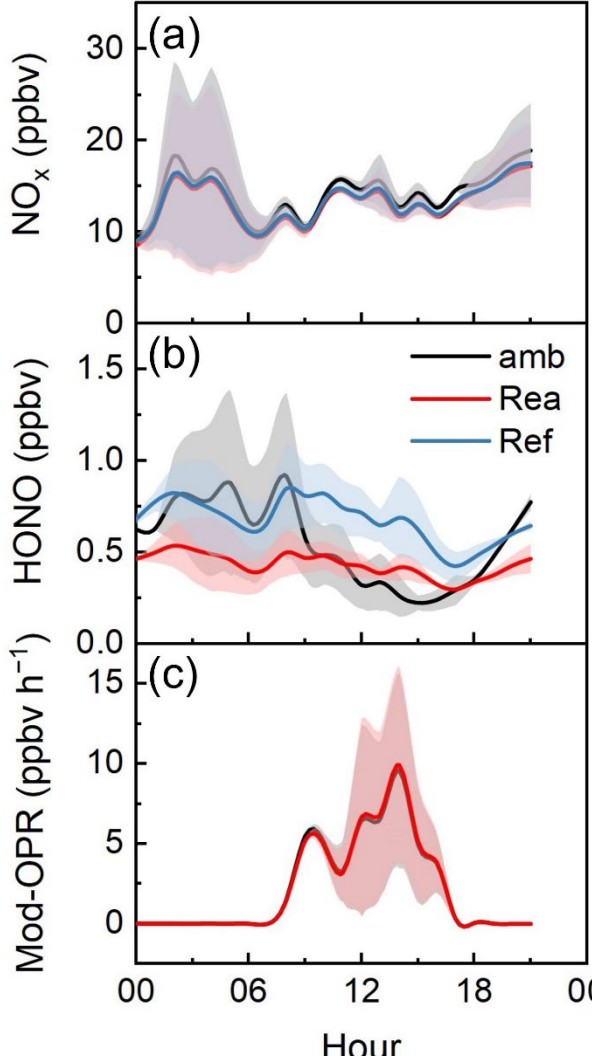


**Figure 2: Diurnal variation of average hourly (a) $NO_x$ and (b) HONO concentrations in ambient air and both chambers. (c)**
**Comparison of $O_3$ production simulation as prescribed with HONO in ambient air and reaction chamber during condition**





**experiment of HONO production. Shaded areas represent 1SD variation of NOₓ, HONO, and OPR for the 1-week HONO production**
**experiment.**

### 3.3 Uncertainty evaluation for Mea-OPR

The detection limit of 2.8 ppbv h$^{-1}$ of our Mea-OPR was determined by three times the standard deviation of Mea-OPR
during nighttime (21:00–04:00) when OPR was supposed to be near null. Currently, tropospheric OPR in remote and rural
areas is believed to be lower than 5 ppbv h$^{-1}$ (Kanaya, 2002; Xue et al., 2013; Bozem et al., 2017). In polluted regions, OPR
can be up to tens of ppbv h$^{-1}$ (Baier et al., 2015; Kleinman, 2005; Xue et al., 2021; Tan et al., 2021; Whalley et al., 2021; Zhou
et al., 2014; Cazorla et al., 2012). Hence, our system is sensitive enough to detect OPR in polluted urban sites, but might not
be so in the clean remote sites.
The uncertainties of Mea-OPR arise from UV transmittance of the chamber wall, measurements of gas residence time and
$\Delta O_x$, and correction from $O_3$ and $NO_2$ wall loss, or HONO production in the reaction chamber (Table S4). Assuming a linear
relationship among Mea-OPR and $j(O^1D)$, reduction in $j(O^1D)$ in the reaction chamber and stray light in the reference chamber
contribute to uncertainties of −4% and −5%, respectively (Table S1). The measurement uncertainty of gas residence time is
evaluated from the multiple residence time measurements to be better than ±0.45%. Evaluated from 1-hour consistency
measurements every night, the measurement uncertainties for $\Delta NO_2$ and $\Delta O_3$ are ±3.8% and ±1.1%, respectively. Higher
uncertainties were found at low levels of $NO_x$ or $O_3$, which suggests that continuous improvement in $\Delta O_x$ measurement
precision will benefit our measurement. The largest uncertainty of Mea-OPR comes from the wall effect of $O_3$ and in our case
the wall effect correction of $O_3$. The wall effect correction functions of $O_3$ indicate a higher wall effect of $O_3$ as $[O_3] \times j(O^1D)$
is higher. As a strong $O_3$ production is also presumed when $[O_3] \times j(O^1D)$ is higher, the wall effect correction of $O_3$ does not
indicate lower measurement quality in strong $O_3$ production scenario. The uncertainties of the fitted $O_3$ uptake coefficient in
both chambers (Eqs. 7–10) are ±4.4% and ±23%, respectively. Wall effect of HONO contributes to an uncertainty of +4.5%,
evaluated from the condition experiment. The total uncertainty of our Mea-OPR can be obtained by error propagation as
described in detail in the Supplement (S2.7). As shown in Fig. S8, the total uncertainty of the Mea-OPR system decreases with
the increase of Mea-OPR. When Mea-OPR is above the detection limit of 2.8 ppbv h$^{-1}$, the uncertainty of Mea-OPR system is
stable and low at an average value of ±27%. In addition, when the $O_3$ uptake coefficient decreases, the uncertainty of Mea-
OPR also clearly reduces, which further indicates the importance of reducing $O_x$ uptake in the chambers for the accurate
measurement of Mea-OPR.

### 3.4 Field application in urban site of Beijing

The mean diurnal profile of ambient $j(O^1D)$, NO, $NO_2$, $O_3$ and Mea-OPR during the Beijing 2022 Winter Olympic Games
are shown in Fig. 3. $NO_x$ levels ranged from approximately 0.5 to 112.8 ppbv, with an averaged value about 18.2 (±16.7) ppbv,
which was similar to that observed at the suburban site (Huairou Station) of Beijing from January to March, 2016 (Tan et al.,
2018b), but much lower than our measurement of 32.0 (±22.4) ppbv in the winter of 2021 (not shown). Specific pollution





control measures for the Beijing 2022 Winter Olympic Games reconcile such a discrepancy. The maximum daily concentration
of $NO_2$ climbed to 26.6 ppbv during the end of our field campaign, which was normal as compared to the winter 2021
campaign. $O_3$ concentration showed afternoon (14:00–16:00) maxima and early morning (6:00–8:00) minima (Fig. 3b), the
former of which was a typical feature of photochemical production. The $O_3$ levels varied from 1.2 to 90.5 ppbv in the daytime,
and the mean and median values were approximately 34.4 (±13.9) ppbv and 36.6 ppbv, respectively. In fact, $O_3$ titration by
NO in this urban site was apparent during the morning/afternoon rush hours, but less evident at noon as it was seen from the
$NO/NO_2$ ratio during rush hours (0.20) and noon (0.46), in comparison with the photo-steady-state ratio of 0.14 and 0.44,
respectively. Considering the $O_3$ titration, the urban diurnal $O_x$ concentration was 43.5 (±3.1) ppbv, higher than the regional
background of 38.0 ppbv (Xu et al., 2020), suggesting the urban area was still the regional source of $O_3$ pollution. Noontime
$j(O^1D)$ peaked at approximately $1.0 \times 10^{-5}$ $s^{-1}$ on average (Fig. 3c), less than half of the value during the summer ($2.6 \times 10^{-5}$
$s^{-1}$) (Tan et al., 2019a). Mea-OPR showed similar diurnal variation to $j(O^1D)$. Mean Mea-OPR peaked at 7.4 ppbv $h^{-1}$ at around
12:30 (Fig. 3c), which was smaller than the $O_3$ production peak of 20 ppbv $h^{-1}$ measured using the Obs-OPR method in summer
at the PKU urban site (Tan et al., 2019a). The nighttime Mea-OPR approached zero as it was also expected from the low levels
of $O_3$, $NO_3$, and $RO_2$ (not shown), accompanied with a nighttime NO of 2.3 ppbv. Therefore, such high Mea-OPR confirmed
active $O_3$ photochemistry and that urban Beijing was still a regional source of $O_3$ pollution even though the pollution control
policy was implemented during the Winter Olympic Games.


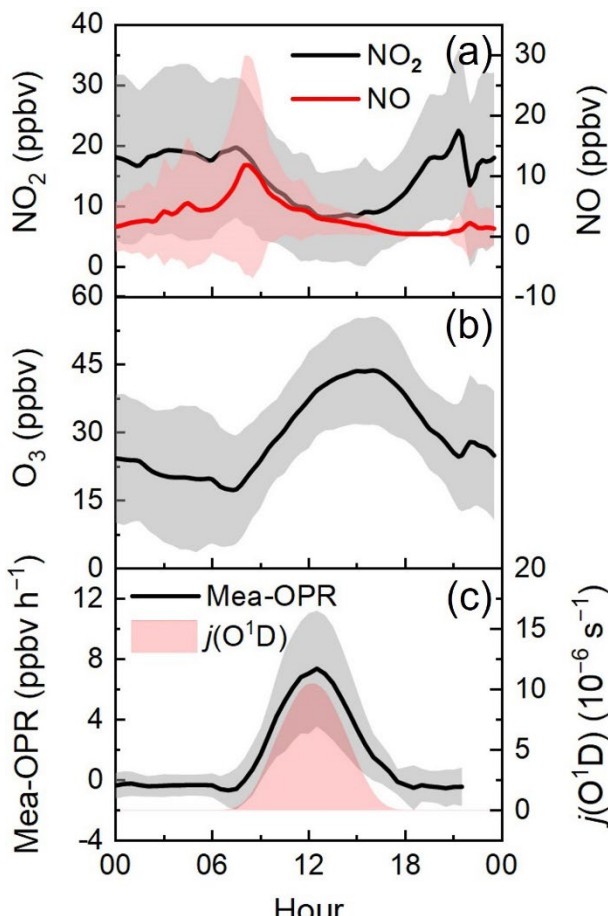

**Figure 3: Mean diel profiles of (a) NO and NO₂, (b) O₃, and (c) Mea-OPR and $j(O^1D)$ at PKU urban site during field campaign. Shaded areas represent 1SD variation of measurement parameters.**

A linear relationship was established for Mea-OPR *versus* $j(O^1D)$, but an atypical relationship was established for Mea-OPR *versus* NO from the Gaussian-shaped ones (Fig. 4) (Whalley et al., 2021; Cazorla et al., 2012). The linear relationship between Mea-OPR and $j(O^1D)$ justified that our Mea-OPR measurement captured its photochemical pattern well. The fitting slope of the Mea-OPR *versus* $j(O^1D)$ plot was $6.1 \times 10^5$ ppbv h$^{-1}$ s$^{-1}$. Compared with a study of Xue et al. (2013) in a global background site of Waliguan, the fitting slope in this study was higher, which might suggest the relatively active nature of O₃ photochemistry in winter in Beijing. With relatively high NO$_x$ level and assumed VOCs-limited regime for O₃ production in Beijing, O₃ photochemistry was expected to be suppressed under high-NO$_x$ conditions. However, an atypical relationship was observed here, as also being validated in Obs-OPR *versus* NO plot (Whalley et al., 2021; Cazorla et al., 2012). The continuous increase in OPR as NO$_x$ increased even in high-NO$_x$ conditions might have accounted for the active O₃ photochemistry in urban Beijing. Detailed reasons for such atypical relationship are not yet clear, but our Mea-OPR shows its promising capacity to capture both expected and atypical patterns of O₃ photochemistry alongside the change of $j(O^1D)$ and NO.

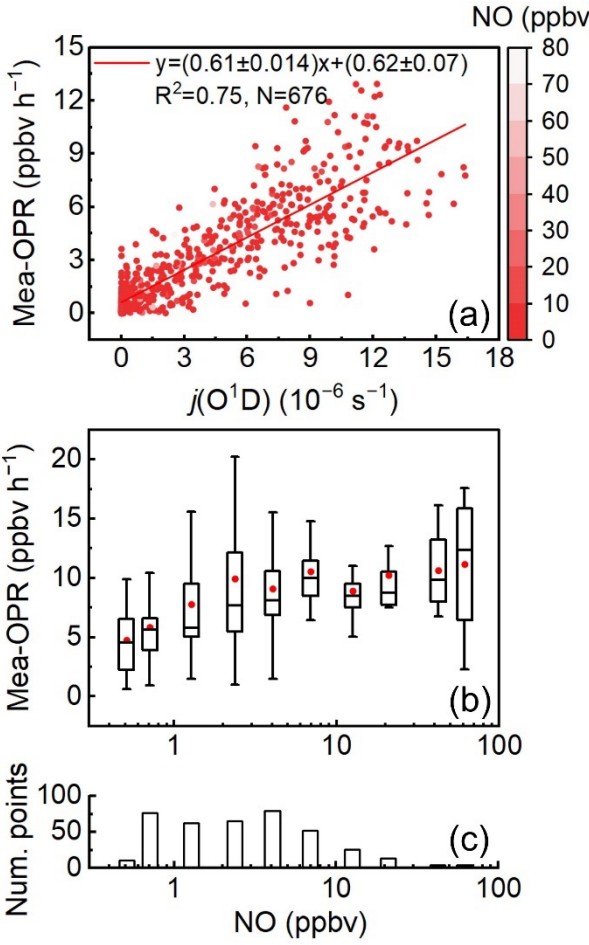

**Figure 4: Plot of Mea-OPR against (a) $j(O^1D)$ and (b) NO for daytime conditions ($j(O^1D) > 10^{-7}$ s$^{-1}$) during field campaign. The interval of NO bin is $\Delta\ln(NO) = 0.57$ ppbv. Panel (c) shows number of datapoints included in each NO interval.**

## 4 Conclusions

Accurate quantification of OPR is an important premise to effective O$_3$ pollution control strategy and to control the adverse effects of O$_3$ pollution on human health and climate. Previous studies have shown discrepancies between Obs-OPR and Mod-OPR, which indicates that our understanding of O$_3$ photochemistry is yet-perfect. Direct measurement of OPR using a twin-chamber system (Mea-OPR) could provide an accurate measurement of OPR, shining light on the emerging conceptual framework of O$_3$ photochemistry. In this work, we reported an optimized system for direct measurement of ozone production rate, Mea-OPR, and its employment at an urban site of Beijing. Our study optimized the chamber design in several ways, i.e., considerably increased the chamber volume and, thus, broadened the optional range of flow rate; employment of a large chamber diameter and the Teflon coating on the inner wall of the chamber to suppress the wall effect. We minimized the chamber surface area to volume ratio to 9.8 m$^{-1}$ and the dark uptake coefficient of O$_3$ to the order of $10^{-9}$. In addition, photo-



enhanced uptake of $O_3$ on deposited particulate matter was found to be the major error source of Mea-OPR. Condition
experiments further revealed a photo-enhanced $O_3$ uptake and recommended a quantitative correction. We finally characterized
a detection limit of 2.8 ppbv $h^{-1}$ and a measurement uncertainty of ±27%, which suggests that Mea-OPR is sensitive enough
to measure OPR in urban or suburban environments. Application of this system in urban Beijing during the Winter Olympic
Games recorded a noontime Mea-OPR of 7.4 (±3.8) ppbv $h^{-1}$ and Mea-OPR *versus* $j(O^1D)$ slope of 6.1 × $10^5$ ppbv $h^{-1}$ $s^{-1}$,
which indicates fairly active $O_3$ photochemistry despite the pollution control policy. The fairly active $O_3$ photochemistry was
mainly assisted by a high abundance of VOCs and $NO_x$, atypically high Mea-OPR even under high-$NO_x$ conditions, but
mediated by relatively weak ultraviolet radiation.

*Data availability.* Contact the corresponding author for data.

*Supplement.* The following file is available free of charge. Optimizing a twin-chamber system for direct ozone production rate
measurement_ SI

*Author Contributions.* Y.W. and C.Y. built the Mea-OPR system, Y.C. and S.C. conducted the condition experiments,
interpreted the data, and wrote the manuscript with revision mainly from Y.W., C.Y., and other authors. Y.W., Y.C., and S.C.
contributed equally to this work and should be considered co-first authors. All authors have given approval to the final version
of the manuscript.

*Competing interests.* The authors declare no competing financial interest.

*Acknowledgment.* We would like to thank all group members in @Beijing 2022 Winter Olympics campaign.

*Financial support.* This work was supported by the National Natural Science Foundation of China (Grants Nos. 41875151 and

408    91744206).












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
