# Peer review of "Optimizing a twin-chamber system for direct ozone production rate"

_Atmospheric Measurement Techniques, 2023_

## Author Comment (AC1)

**Response to reviewers' comments**

RC1:

This manuscript reports on the development of an instrument for measuring ozone production rates in the atmosphere. While this type of instrument was already published in the literature, the authors present an interesting approach to account for wall effects that were found to significantly impact the measurement accuracy . Such developments are scarce for this technique and this work is of interest for the scientific community.

However, the writing must be revised. It was not easy to read this publication, the meaning of a large number of sentences being difficult to understand. A few examples are provided below but there are other instances in the text. In addition, the authors should address the specific comments indicated below.

This reviewer recommends major revisions with a second round of reviews.

Response: Many thanks for revision suggestions. We have conducted a major revision to clarify key technique details and improve writing clarity. A point-to-point response is also prepared.

Examples of confusing words/terminology

- L35: changed to "imperfect"
- L38 & L188: changed to "control experiments"
- L65: changed to "photochemical production"
- L102 changed to "box model constrained with observations"
- L165: changed to "parallel measurements"
- L221: "constrained with comprehensive measurement of parameters concerning $O_3$ photochemistry"

Examples of sentences that are difficult to read, unclear, which need to be rephrased

- L43-44: "… fairly active $O_3$ photochemistry, which was assisted by a high abundance of VOCs and $NO_x$, atypically high Mea-OPR even under high-$NO_x$ conditions, but mediated by relatively weak ultraviolet (UV) radiation."
- L125-126: "Measurement uncertainties in $O_x$ might be enlarged as transferred to be the measurement uncertainties in $\Delta O_x$, thus further amplifying the measurement uncertainties of Mea-OPR in a unit of ppbv $h^{-1}$ because the gas residence time in Mea-OPR chambers is typically much shorter than 1 hour."
- Entire section 2.2

Response: Revisions are shown as follows:

- L35: changed to "imperfect"
- L38 & L286: changed to "control experiments"
- L66: changed to "photochemical production"
- L107: changed to "box model constrained with observations"
- L253: changed to "parallel measurements"
- L368: "constrained with comprehensive measurements of parameters concerning $O_3$ photochemistry"
- L43-44: The sentence is deleted.
- L125-126: Please refer to revision in lines 137-139.
- Entire section 2.2: Please refer to revision in lines 286-388.

Specific comments:

- L 146-148: The authors discuss how $O_x$ losses change between raw quartz material and Teflon coated material. The authors should provide quantitative information here, referencing Table S3 from the supplementary material.

Response: Suggestions are followed. Please refer to our revision in lines 164-165 "Corresponding uptake coefficient of $O_3$, $\gamma_{O3}$, is calculated in this context to be $5.2 \times 10^{-9}$ on Teflon wall surface and $7.1 \times 10^{-8}$ on quartz surface."

- L149 Eq. 4: Please define the parameters in the text.

Response: Please refer to our revision in lines 167-169 "where $\Delta O_3$, uptake is the differential $O_3$ between the ambient and the chamber due to uptake loss of $O_3$ in the chamber. $O_{3, amb}$ is the ambient $O_3$ concentration in ppbv. $\omega_{O3}$ represents the mean molecular velocity of $O_3$, in m s$^{-1}$. $\tau$ is the mean gas residence time in the chamber in second."

- L150: "To date, these effective designs have not yet been integrated and evaluated in a state-of-the-art version of Mea-OPR" – The authors should clarify what is meant here by "effective designs".

Response: We added in lines 171-172 "All in all, previous studies recommend wall materials of high light transmittance for the reaction chamber, precise and stable instrument to measure $\Delta O_x$, and suppression of wall effect by employing inert material and large diameter of the chamber."

- Section 2.1: The authors should provide more details on this new OPR system. How is ambient air introduced into the chambers? What type of inlet? What type of Teflon coating (brand)? What type of UV filter for the reference chamber? How is the sampling performed from the chambers? What type of outlets? It seems from Figure S1 that some air is also extracted from the chambers using MFCs and a pump. How is it done? Sampling flow rates from $O_3$ and $NO_2$ monitors?

Response: We have rewritten the experiment section to provide more details. Please refer to lines 179-285.

- L171-172: "The instrument alignment experiment suggested that nigttime $\Delta O_x$ was 0.07 ($\pm 0.26$) ppbv, within the instrument detection limits (Fig. S5)." – How did the authors get the number of 0.07 ppb? Is it an average value for the whole time series? If so, it should be clarified. Looking at Fig. S5, it is clear that $\Delta O_3$ and $\Delta NO_2$ display a similar increasing trend over the whole time period (total increase of approximately 0.4 ppb). Is it due to a drift in the monitors' zero? If so, why is it similar for both types of monitors?

Response: The increasing trend is possibly a result of small differences between two adjacent calibrations of $O_3$ analyzer and $NO_x$ analyzer or drifts of instrument zero. Nighttime $\Delta O_x$ is an average value for the whole time series. Please refer to our revision in lines 259-260 "The campaign average of $\Delta O_x$ is 0.07 ($\pm 0.26$) ppbv in fact within the instrument detection limits of 1.0 ppbv for $O_3$ analyzer."

- L178-180: "Another key design to suppress the wall effect was the transparent Teflon coating, which was essential for accurate Mea-OPR by reducing the uptake coefficient of O3 from $10^{-8}$ on quartz wall (Sklaveniti et al., 2018) to $10^{-9}$ on Teflon coating wall under dark conditions." – The authors should also address how the uptake of $NO_2$ changes between quartz and Teflon. It is likely that the $NO_2$ uptake is larger for Teflon since this material is more hydrophilic than quartz.

Response: Great idea! The revison is shown in lines 199-202 "In addition, slight suppression on uptake coefficient of $NO_2$ ($\gamma_{NO2}$) of $6.3 \times 10^{-8}$ on Teflon coating wall in the reference chamber for our system under dark conditions relative to report value of $7.0 \times 10^{-8}$ on quartz wall is also evidential (Sklaveniti et al., 2018; Sadanaga et al., 2017)."

- L187 Eq. 5: This equation must be demonstrated and the authors should add a section in the supplementary material to present how they derived it. Is "$\varphi_{trans}$" for the reaction chamber only? If so, please clarify it in the text. What are the assumptions made to derive this equation? It seems that the authors consider that Mea-OPR scales linearly with "$\varphi(trans)$" and "$\varphi(\Delta HONO \text{ or } \Delta NO_x)$". The authors should discuss the validity of these assumptions. This equation also deserves more discussion in the main paper to highlight how wall losses of $O_3$, UV transmission and surface production of HONO are corrected for. Why did the authors decided to not include a correction for $NO_2$ wall losses?

Response: Indeed! We added a paragraph to explain our measurement definition and correction of Mea-OPR. Revisions in lines 266-285 are shown.

- Section 2.2 : This section needs major revisions. This reviewer had difficulties to understand what was done here. In addition, indicating that the production of HONO is due to the heterogeneous hydrolysis of $NO_2$ is too restrictive. Light-induced processes leading to the conversion of $NO_2$ into HONO at the chamber's surface should be discussed.

Response: Please refer to our revision in line 336 "Uptake loss of $NO_2$ and HONO production from $NO_2$ uptake (R1) or unknown sources on the Teflon film and quartz surface" and lines 513-515 "Daytime source of HONO in zero-OPR control experiment has suggested that this source depended on chamber contamination rather than $NO_2$ concentration. Both heterogeneous uptake of $NO_2$ and unknown source of HONO might account for the daytime HONO production."

L220: The authors mention that "MCM model was conducted to calculate $O_3$ production in chambers". However, there is no information about the model used in this work. The authors should add a section in the supplementary material to provide details about the model, the chemical mechanism, and how the model was constrained.

Response: We added a paragraph to describe the model construction and constraint. Please refer to our revision in lines 377-388.

- L228-231: "Under typical working conditions of Mea-OPR, $O_3$ uptake loss contributes to a false Mea-OPR signal of 20.3 ppbv $h^{-1}$ at uptake coefficient of $7.11 \times 10^{-8}$ and S/V ratio of 18 $m^{-1}$ (the least in the literature), relative to a false Mea-OPR signal of 1.29 ppbv $h^{-1}$ at uptake coefficient of $8.12 \times 10^{-9}$ and S/V ratio of 9.76 $m^{-1}$ in our Mea-OPR, assuming ambient $O_3$ concentration of 50 ppbv." – This comparison does not seem pertinent. When Sklaveniti et al. report a potential bias of approximately 20 ppb/h at an ozone mixing ratio of 50 ppb, this is for daytime conditions when the photo-enhanced loss of ozone is operating in the reaction chamber. For the present instrument, the uptake coefficient taken into consideration is for dark conditions. As the authors indicate on L234-235, the $O_3$ uptake coefficient for daytime conditions is approximately one order of magnitude larger, which would lead to a "false Mea-OPR signal" of 12.9 ppb/h. So, while an improvement is indeed observed, the magnitude of this improvement is not as large as stated.

Response: We have added a Table 1 to compare our results with previous reports. Also, measurement bias associated with $O_3$ uptake for our Mea-OPR in nighttime and daytime are comprehensive discussed in lines 464-468 "It is the difference in $\gamma_{O3}$ between the two chambers that brings bias for $\Delta O_x$ measurements. The difference in $\gamma_{O3}$ of $4.6 \times 10^{-9}$ ($\pm 2.0 \times 10^{-9}$) is ca. one third of $\gamma_{O3}$ in the reaction chamber under dark conditions. $\Delta O_x$ measurement bias of 0.78 ($\pm 0.85$) ppbv $h^{-1}$ is then calculated for our Mea-OPR during our field campaign. Correction of this measurement bias brings Mea-OPR of $-0.46$ ($\pm 0.75$) ppbv $h^{-1}$ to 0.31 ($\pm 0.92$) ppbv $h^{-1}$ in the nighttime during the field campaign." and lines 482-494.

- L239-241: "A routine water flush cleaning and UV-photochemical-aging cleaning of both chambers were then scheduled after occurrence of heavy pollution episodes" – Nothing is said about the effectiveness of these cleaning periods. The authors should discuss whether these were useful to reduce the ozone uptake on the chambers' wall.

Response: So far, we have conducted 5 zero-OPR control experiments and have measured uptake coefficient from $8.0 \times 10^{-8}$ to $4.0 \times 10^{-7}$ under the $j(O^1D)$ of $1.0 \times 10^{-5}$ $s^{-1}$. The high uptake coefficient is collected after heavy pollution episodes while the lowest uptake coefficient is collected after our cleaning procedures. Our cleaning procedures are very effective in

suppression of $O_3$ uptake loss. The zero-OPR control experiments on date 14-16 November, 2023 (before cleaning) and 30 November-2 December, 2023 (after cleaning) are shown in Fig. R1 as an example.

[Figure]

Figure R1: The relationship between $\gamma_{O_3}$ and $j(O^1D)$ during two zero-OPR control experiments on 14-16 November, 2023 (before) and 30 November-2 December, 2023 (after), respectively. "Before" represent the zero-OPR control experiment conducted before cleaning. "After" is the zero-OPR control experiment conducted after cleaning. It can be seen that the uptake coefficient of $O_3$ is reduced by half after cleaning.

- L258-260: "After multiple control experiments (not shown), we could also assume the uptake coefficient of $O_3$ being stable between two adjacent control experiments." – These results are important to ensure that the correction parameterized through Eqs. 7-10 is suitable for the whole campaign. The authors should show and discuss these additional experiments in the supplementary material.

Response: We conducted two adjacent zero-OPR control experiments on 5-6 January, 2023 and 14-15 March, 2023. It can be seen from Fig. R2 that the uptake coefficient of $O_3$ barely changes during two-month field campaign. Thus, we could assume the uptake coefficient of $O_3$ being stable in our field campaign from 5 February to 3 March, 2022. In general, we recommend at least one zero-OPR control experiment for a four-week field campaign and chamber cleaning after heavy pollution episodes or before a new field campaign.

[Figure]

Figure R2: The relationship between $\gamma_{O_3}$ and $j(O^1D)$ in two adjacent zero-OPR control experiments (5-6 January, 2023 and 14-15 March, 2023). It can be seen that the uptake coefficient of O3 barely changes during two-month field campaign.

- L261-263: Eqs. 7-8 do not account for the potential impact of RH on the O3 uptake. The reviewer understands that deriving the RH-dependence during daytime is challenging due to fast changes in $j(O^1D)$. However, since several "control experiments" were performed, wouldn't it be possible to group all the results to investigate the RH-dependence within bins of J-values?

Response: Great idea. We calculate the fitting residual by subtracting the data points to the fitting line. The fitting residual shows no dependence on RH in daytime (Figure R3). Therefore, multiple regression fitting for uptake coefficient concerning its dependence on $j(O^1D)$ and RH isn't done.

[Figure]

Figure R3: Plot of fitting relative residual in Fig 3c and RH during zero-OPR control experiments (5–6 February, 2022). Uptake coefficient- $j(O^1D)$ fitting shows a high uncertainty in low $j(O^1D)$ conditions. However, fitting residual is randomly distributed, rather than depends on RH.

- L269-271: "Despite of considerable uptake coefficient of $NO_2$, compared with that of $O_3$, much lower $NO_2$ level (12.9 ppbv) relative to $O_3$ inferred negligible wall loss of $NO_2$, which was transferred to ca. 1.79 ppbv h−1 false signal of Mea-OPR at most during the 1-week HONO production experiment." – Do the authors mean that the loss of $NO_2$ at the surface leads to a negligible loss of $O_x$ species? If so, this should be clarified. In addition, the stated bias is similar, even larger, than that reported for the dark $O_3$ wall loss on L230. So, how could it be negligible? Do the authors mean that this is negligible compared to the light-induced $O_3$ wall loss?

Response: This has been clarified in lines 496-502 "In the 1-week HONO production experiment, net $NO_x$ uptake loss reached 1.00 ($\pm$0.65) and 0.76 ($\pm$0.65) ppbv in the reaction and reference chamber, relative to the ambient (Fig. 5a). Uptake coefficient of $NO_2$ in the reaction chamber and reference chamber are calculated $8.3 \times 10^{-8}$ and $6.3 \times 10^{-8}$, which is actually comparable to daytime uptake of $O_3$ and slightly less than previous measurement of $NO_2$ uptake on quartz chamber of $7.0 \times 10^{-8}$ (Sklaveniti et al., 2018). Due to the lack of light-dependence, differential $NO_2$ uptake between the two chambers is much less, compared with differential $O_3$ uptake. Much lower $NO_2$ level (14.7 ppbv vs 28.7 ppbv) than $O_3$ during our field campaign also rationalizes much lower measurement bias associated with $NO_2$ uptake. Eventually, measurement bias of 0.72 ppbv $h^{-1}$ for Mea-OPR is calculated."

- L279-280: "RH in the reaction chamber scattered at approximately 61% ($\pm$14%) and was much higher than the ambient air of 36% ($\pm$14%) during the nighttime" – How do the authors explain that RH in the reaction chamber could be significantly larger than in ambient air?

Response: This has been corrected and clarified in Figure S3.

- L281-282: The authors should consider to show how the HONO uptake varies during the night as they did for O3 in Fig. 1b. This could be included in the supplementary material. Did the authors investigate whether the HONO uptake depends on environmental variables such as T and RH?

Response: We agree with the reviewer that uptake and releasing of HONO is interesting. As seen Fig. R4, HONO uptake depends on its concentration, reflecting a partitioning equilibrium among HONO and particulate nitrite deposited on chamber wall. This result will be further summarized in a separate manuscript.

[Figure]

Figure R4: Plot of HONO production, P(HONO), in the reaction chamber againest (a) $j(O^1D)$ and (b) HONO measurements in the ambient.

- L304-305: "Currently, $NO_2$ uptake and HONO production correction are not applied for our Mea-OPR." – If corrections are not applied, the term "$\varphi(\Delta HONO\ or\ \Delta NOx)$" should be removed from Eq. 5.

Response: We take the advices of reviewers to include all the corrections in Eq. 5. We have added a new paragraph to state our measurement definition of Mea-OPR in lines 266-285.

- L322-323: "Evaluated from 1-hour consistency measurements every night, the measurement uncertainties for $\Delta NO_2$ and $\Delta O_3$ are $\pm3.8\%$ and $\pm1.1\%$" – Do these errors only account for drifts in monitors' zero? Or do they factor other sources of uncertainties such as errors associated to the calibrations reported in Table S2 and the concentration of the calibration gases?

Response: Yes, both shift in responsing sensitivity and instrument zero might contribute to the measurement uncertainties for $\Delta NO_2$ and $\Delta O_3$. Higher uncertainties at low levels of $NO_x$ and $O_3$ also suggests that baseline shift might be mainly responsible for this measurement uncertainty.

We have revised in line 559-561 "Higher uncertainties are found at low levels of $NO_x$ or $O_3$, which suggests that continuous improvement in $\Delta O_x$ measurement precision will benefit our measurement."

- Section 3.4: The authors should consider showing and discussing the entire OPR time series in addition to the mean diel profile. Discussing the day-to-day variability of OPR and $NO_x$ would nicely complement Fig. 4.

Response: We agree with reviewer on more detailed discussion on Mea-OPR and its response with precursor concentrations, meterological parameters, etc. In another prepared manuscript, we compare our measurements in two cities (Beijing and Lhasa), and shows distinct $O_3$ photochemistry in these two urban environments. OVOCs or OVOCs/$NO_x$ ratio in the two cities appears to account for distinct $O_3$ photochemistry. With the restriction of manuscript lengthen and writing scope, we only choose to show the potential of our Mea-OPR system to characterize $O_3$ photochemistry in urban environments.

- Figure 3: The authors should add an additional panel to show ozone production rates that would be calculated when the $O_3$ wall loss is not corrected for. What is the magnitude of the correction?

Response: The corrections associated with $O_3$ wall loss is essential for Mea-OPR. This also identifies the major uncertainties of Mea-OPR as shown (Fig. 4 in the context). Similarly, the corrections associated with $NO_2$ wall loss is also a considerable part of Mea-OPR.

- Figure 4: The reviewer recommends using another color coding for NO. The datapoints close to 80 ppb NO are not visible.

Response: Suggestions are accepted.

- The recent publication from Morino et al. (Atmos. Environ., 309, 2023) is not referenced and the authors may want to include it.

Response: Suggestions are accepted.

**Edits:**

- L118: "… or named MOPS in literature" should read ""… or named MOPS or OPR instrument in the literature"

- L135-136: "Sklaveniti et al. (2018) estimated that $O_3$ uptake loss in their version of Mea-OPR would lead to false Mea-OPR signal of ~20 ppbv h$^{-1}$ assuming ambient $O_3$ to be 50 ppbv." Should read "Sklaveniti et al. (2018) estimated that a photo-enhanced $O_3$ uptake in their version of Mea-OPR would lead to false Mea-OPR signal of ~20 ppbv h−1 assuming ambient $O_3$ to be 50 ppbv."

- L141: "is usually chosen for it is easier to manipulate" should read "is usually chosen since it is easier to manipulate"

- L177-178: "… while enabling multiple sampling instruments equipped" should read "while enabling the sampling from several instruments"

- L321: "contribute to uncertainties of −4% and −5%, respectively" should read "contribute to a systematic bias of −4% and −5%, respectively

Response: Suggestions are accepted.

---

## Author Comment (AC2)

**Response to reviewers' comments**

RC2:

The manuscript "Optimizing a twin-chamber system for direct ozone production rate measurement" by Wang et al. presents an instrument designed to measure directly the total in-situ ozone production. This is indeed a measurement that could be useful to improve our understanding of ozone formation. Instruments such as this have been described in the past, and deployed in the field with varying degrees of success. The authors claim to have substantially improved the technique over previous designs and to be able to achieve a detection limit of 2.8 ppb/h with a 27% uncertainty. This would be a great development, but I don't see in this manuscript much evidence to back the authors' claims, to be honest.

For the most part, the paper is written rather confusingly. The description of the characterization experiments is severely lacking details, and the information provided is limited to 1 day or diurnal averages. This is not sufficient to allow proper evaluation of the instrument's performance or the author's claims. Additionally, the characterization experiments are not described properly. The text and the figures suggest that the instrument was sampling ambient air for these experiments which would not be a good method to characterize a new instrument. One would want to do this type of experiments under controlled conditions, with air of known composition, especially if the goal is to demonstrate improved performance. At the moment, I cannot recommend publication because I see several serious methodological errors that undermine the authors' claims.

Response: Many thanks for the suggestions. We have revised the manuscript accordingly and included key content originally in supplementary material to clarify these points mentioned by reviewers. A point-to-point response is also presented.

Specific comments
* * *
In section 3.1, it is said that during the zero $NO_x$ high $O_3$ experiments the wall loss was more than 3x higher in the reaction chamber than in the reference chamber. This seems a pretty significant factor to me that could have large impact on the instrument's performance. I think the authors should elaborate on the possible causes (is it the teflon film? differences in humidity?), and also comment on the effect on the measurements. Was this difference constant through the measurement period? Impossible to say from the data presented here.

Response: The dependence of $O_3$ wall loss on $j(O^1D)$ suggests that photo-enhanced uptake of $O_3$ on chamber wall or deposited aerosol on chamber wall might account for differential uptake between the two chambers. The fitting between uptake coefficient- $j(O^1D)$ is found to be well, with a fitting residual larger than 12.7% of $\gamma_{O3}$ in the morning and dawn, but less than 2.0% of

$\gamma_{O3}$ during the noontime. The fitting residual shows no dependence on RH in zero-OPR control experiments (Fig. R1). Please refer to our revision in lines 454-458.

[Figure]

Figure R1: Plot of fitting relative residual in Fig 3c and RH during zero-OPR control experiments (5–6 February, 2022). Uptake coefficient- $j(O^1D)$ fitting shows a high uncertainty in low $j(O^1D)$ conditions. However, fitting residual is randomly distributed, rather than depends on RH.

We conducted two adjacent zero-OPR control experiments on 5-6 January, 2023 and 14-15 March, 2023. It can be seen from Fig. R2 that the uptake coefficient of O₃ barely changes during two-month field campaign. Thus, we could assume the uptake coefficient of O₃ being stable in our field campaign from 5 February to 3 March, 2022.

[Figure]

Figure R2: The relationship between $\gamma_{O_3}$ and $j(O^1D)$ in two adjacent zero-OPR control experiments (5-6 January, 2023 and 14-15 March, 2023). It can be seen that the uptake coefficient of O₃ barely changes during two-month field campaign.

A "routine water flush" to eliminate particles from the chamber is mentioned on page 8. First of all, if this is only done after "severe pollution episodes" it is not routine and, second, what were the criteria to decide when was it needed? More importantly, I would expect this procedure to have an important impact on the wall interactions of $O_x$. Presumably, it will lead to higher presence of water on the surfaces and therefore more issues with the wall loss and/or HONO production.

Response: Please refer to our revision in lines 440-444 "A water flush cleaning and UV-photochemical-aging cleaning of both chambers are then scheduled before new field campaign or after occurrence of heavy pollution episodes ($PM_{2.5} > 80$ µg cm$^{-3}$), following the recommendation in previous literature (Chu et al., 2022). Water flush cleaning is found effective to remove deposited aerosol particles on chamber wall. UV-photochemical-aging cleaning not only dry the chamber, also deactivated the wall surface. The two-step cleaning process is found to effectively reduce wall loss of $O_x$ to a lower rate of this report (not shown)."

Our cleaning procedures are very effective in suppression of $O_3$ uptake loss. The zero-OPR control experiments on date 14-16 November, 2023 (before cleaning) and 30 November-2 December, 2023 (after cleaning) are shown in Fig. R3 as an example.

[Figure]

Figure R3: The relationship ship between $\gamma_{O_3}$ and $j(O^1D)$ during two zero-OPR control experiments on 14-16 November, 2023 (before) and 30 November-2 December, 2023 (after), respectively. "Before" represents the zero-OPR control experiment conducted before cleaning. "After" is the zero-OPR control experiment conducted after cleaning. It can be seen that the uptake coefficient of $O_3$ is reduced by half after cleaning.

Overall, it is not possible to assess the authors' claim that their procedures lead to improved performance of the instrument based on just one day of observations shown here, especially since they look like ambient observations.

Response: We clarified the design and data analysis zero-OPR control experiment in the revised manuscript. Please refer to lines 288-334 "As Mea-OPR system samples the ambient air, wall loss and photochemical production of $O_3$ are occurring simultaneously. Therefore, these two processes cannot be decoupled from each other. A control experiment sampling diluted $O_3$ standard gas (generated from zero air passing through ozone generator) has been designed and

conducted. $O_3$ is diluted to around 113 ppbv by zero air before entering the chambers. High $O_3$ concentration in zero-OPR experiments facilitates measurements of $O_3$ wall loss. Due to extremely low $NO_x$ and low VOCs in the zero-air supply, suppressed $O_3$ photochemistry, apart from $O_3$ photolysis, in the reaction chamber is assumed. The control experiment is thus referred as zero-OPR control experiment. Zero-OPR experiments have been conducted for several field campaigns so far. Other zero-OPR control experiments measure changes of $O_3$ uptake in winter when heavy haze, i.e. $PM_{2.5} > 80$ μg m$^{-3}$ (Chu et al., 2022), occurs frequently in Beijing (not shown). This might lead to the contamination of the chamber (Sklaveniti et al., 2018). Therefore, we recommend each zero-OPR control experiment at least during one field campaign to check the $O_3$ wall loss. Herein, results from zero-OPR control experiment conducted on 5-6 March, 2022 during the first employment of our Mea-OPR in Beijing are shown as an example in the context. Before the $O_3$ enters both chambers, another $O_3$ analyzer monitors the diluted $O_3$ standard gas, referred as $O_{3,\ amb}$. An excess flow rate of 1.0 L min$^{-1}$ is to maintain 1 bar pressure in the quartz chamber. $O_3$ travels through chambers and is then sampled via the main outlets to measure $O_3$ concentration in the two chambers. Measurements of $O_3$, $NO_2$, NO (Thermo Scientific, Model 42i, LOD: 0.4 ppbv), HONO and CO (Thermo Scientific, Model 48i, LOD: 0.04 ppmv) in chambers have been simultaneously conducted in control experiments (Fig. 2). The measurements of other species are to check the experiment control of $O_3$ precursors in zero-OPR control experiments……"

I don't understand this sentence on page 8: "The MCM model was conducted to calculate $O_3$ production in chambers". There has been no mention of a model in previous pages. How was the model constructed and run? It seems that the model was constrained to $NO_x$ and CO (or just initialized, please clarify what is meant with "prescribed"), but there is no information on the other parameters: VOCs, humidity, photolysis rates etc... Some of this information is in the supporting information but it should be at least referred to.

Response: We have moved the method section originally in the supplementary material into the main context of our manuscript. We have also added a new paragraph to describe model construction and model constraint. Please refer to revision in lines 377-388 "Mechanisms are extracted from the website of Leeds University (MCM v3.3.1, http://mcm.leeds.ac.uk/MCM) for our chemical model to mimic the oxidation of VOCs and inorganic species in the chamber. Uptake of $O_3$ or uptake of $NO_2$ or HONO production are not included in the model as $O_3$, $NO_x$ and HONO are constrained with our measurements. For simulation of zero-OPR control experiments in Model S0, only oxidation of CO is calculated by the model, as oxidation of VOCs are not expected in the zero-OPR control experiments. For simulation of 1-week HONO production experiments in Model S1–S3, oxidations of measured VOCs and model-generated photochemical intermediates are calculated. The preliminary model run (not shown) suggests that $k_{OH}$ contributed by $NO_x$, CO, VOCs, OVOCs, and model-generated intermediates underestimates the measured $k_{OH}$. This indicates a missing $k_{OH}$ as compared with $k_{OH}$ measurements, which has been described in Wei et al.(2020). Additional formaldehyde (HCHO) and HCHO + OH reaction is then included in chemical model to represent the missing $k_{OH}$ and better represent $O_3$ chemistry (Tan et al., 2021). Model constraints include measurements of HONO, $NO_x$, $O_3$, CO, VOCs, OVOCs, $j$ values, T, RH, etc."

It appears (line 225) that the model was used to calculate the photochemical production of $O_3$ and to estimate the $O_3$ wall loss by subtracting it from the instrument's output. First of all, it is not clear at all that this is what has been done, and the authors should describe the procedure more accurately. But, if this is the case, I see a major issue with this procedure because it relies on the assumption that an MCM model can predict $O_3$ production with great accuracy. This is most likely not true, especially under ambient conditions (which the following paragraph, lines 234-246, suggest was the case), and in particular it will not be true with the level of accuracy that would be required to estimate the uptake coefficients in the chambers to the degree that the authors claim. Using a model in this way undermines the whole discussion on $O_3$ uptake in Section 3.1.

Response: We have clarified this issue in lines 414-428 "As shown in Fig. 3, evident $O_3$ uptake loss was observed in both chambers and higher $O_3$ uptake loss was observed in the reaction chamber ($\Delta O_3$ = 7.7 ppbv) relative to the reference chamber ($\Delta O_3$ = 2.3 ppbv) at noon. $NO_x$ concentration was measured around 0.03–1.05 ppbv in the zero-OPR control experiment. A slight increase in $NO_x$ from the morning to the noon is accompanying increasing $j(O^1D)$. Also, a stable and low concentration of $NO_x$ in the zero air before entering the chamber further confirms our attribution of this bridge-shaped $NO_x$ to previously-identified unknown source of HONO and $NO_x$ in Teflon chamber (Rohrer et al., 2005; Akimoto et al., 1987; Ye et al., 2016). MCM model is then conducted to calculate $O_3$ production, $OPR_{zero}$, in both chambers for the zero-OPR control experiment. MCM model is believed to be able to well represent $O_3$ photochemistry in the relatively simple chemical reaction system involving mainly oxidation of CO. $OPR_{zero}$ in the reference chamber is calculated near zero at noon because stray light in the reference chamber was too weak to be meaningful for $O_3$ photochemistry. In addition, abundance of $O_3$ chemical precursors in reference chamber during the zero-OPR control experiments are relatively low. Therefore, $O_3$ loss in the reference chamber is not corrected. $OPR_{zero}$ in the reaction chamber is calculated to be up to 9.0 ($\pm$1.5) ppbv $h^{-1}$ at noon (Fig. S6). $\Delta O_{3,\,uptake}$ in the reaction chamber is therefore corrected for non-zero $OPR_{zero}$ in the reaction chamber and the correction comprised 28% of $\Delta O_{3,\,uptake}$. As $OPR_{zero}$ is much less than $\Delta O_{3,\,uptake}$, $O_3$ photochemical production is still considered to be successfully controlled in the zero-OPR control experiment."

For MCM model, we have clarified this issue in lines 329-332 "Simulations of ambient OPR are somewhat suffering from uncertainties tied to for example imperfect understanding on oxidation mechanism of complex NHMC (Saunders et al., 2003; Hao et al., 2023). Notably, $OPR_{zero}$ simulations by our chemical model are more trustful, relative to simulations of ambient OPR, as simple $O_3$ photochemistry involving only oxidation of CO, but not complex NHMC, is of concerns in the zero-OPR control experiment."

In section 3.2, the authors discuss the formation of HONO in the instrument chambers. Again it is not clear what was done and how during this "1 week HONO experiment". It is also not clear whether HONO was measured or calculated, and how. If the numbers on HONO production cited on page 10 come from the MCM model (which parametrization/reaction scheme?), then I have serious doubts on their reliability.

Response: Please refer to our revision in lines 336-388 for a more detailed description of 1-week HONO experiment and data processing procedure. "Uptake loss of $NO_2$ and HONO production from $NO_2$ uptake (R1) or unknown sources on the Teflon film and quartz surface have been proposed to be error source of Mea-OPR by Baier et al. (2015) and Sklaveniti et al. (2018). Similar to $O_3$ uptake, uptake of $NO_2$ is a potential measurement bias of $\Delta O_x$. Moreover, wall loss of $NO_x$ and production of HONO specially in the reaction chambers perturb $O_3$ photochemistry therein. To obtain uptake loss of $NO_x$ and production of HONO in both chambers, additional measurements of HONO and $NO_x$ in the ambient and in the chambers has been conducted for one week during 10–18 February, 2022. HONO was measured by customized LOPAP. The detailed description of customized LOPAP can be found in Wang et al. (2023). Three sets of identical HONO instruments sample ambient air and chamber air simultaneously. Measurements of chamber HONO allow us to calculate differential HONO between chambers and differential HONO between chambers and the ambient (not shown). This control experiment to characterize $NO_2$ uptake and HONO production in the reaction is conducted during the field application of Mea-OPR. To discriminate it from the month-long field campaign, we refer this control experiment as 1-week HONO production experiment in the context……"

In any case, I disagree very much with their conclusion that this is not an important factor. The authors estimate an uptake coefficient for $NO_2$ of the order of 10^-8 (line 268), which is of the same magnitude as their estimate of $O_3$ uptake coefficient (figure 1). Therefore I think the claim that the wall loss of $NO_2$ (which likely leads to HONO formation) is negligible does not hold. It is also quite apparent from figure 2 that while the $NO_x$ levels in both chambers are similar, HONO levels are not which strongly suggest there is formation in one chamber.

Response: We agree with the reviewer and include these corrections in Mea-OPR (Eq. 5) in the revised manuscript. We have further clarified this point in lines 266-282 "Mea-OPR can be calculated in Eq. (5), which is modified from previous scheme in the literature (Cazorla and Brune, 2010; Sadanaga et al., 2017; Sklaveniti et al., 2018). $\Delta NO_2$ and $\Delta O_3$ are the differential $NO_2$ and $O_3$ between the two chambers, respectively. D is the diameter of chambers in m. $O_{3, amb}$ and $NO_{2, amb}$ represent the ambient $O_3$ and $NO_2$ concentration in ppbv, respectively. $\varphi_{trans}$ is the ratio of in-chamber $j(O^1D)$ to ambient $j(O^1D)$ as determined by the UV transmittance of the two Mea-OPR chambers. We assume linear dependence of OPR on $j(O^1D)$ (Tan et al., 2018a), and therefore Mea-OPR underestimation on ambient OPR associated with chamber filter of UV in the reaction chamber and stray light in the reference chamber can be corrected by $\varphi_{trans}$. Positive $\Delta HONO$ in the daytime is found in our field campaign due to $NO_2$ uptake or unknown chamber source of HONO (Rohrer et al., 2005; Akimoto et al., 1987; Ye et al., 2016). As a result, Mea-OPR tends to be overestimated with a correction factor of $\varphi_{\Delta HONO}$ typically higher than unit. Negative $\Delta NO_x$ is observed in our field campaign due to uptake loss of $NO_2$ on chamber wall. This leads to bias in $\Delta O_x$ measurements and also reaction chamber underestimation of ambient $O_3$ production. As a result, Mea-OPR tend to underestimate OPR with a correction factor of $\varphi_{\Delta NO_x}$ typically lower than unit. In addition, chamber formation of HONO is somewhat associated with uptake loss of $NO_2$, therefore $\varphi_{\Delta HONO}$ and $\varphi_{\Delta NO_x}$ are evaluated together. The overall correction factor associated with imperfect chamber mimic of ambient $O_3$ photochemistry

due to $\Delta$HONO and $\Delta$NO$_x$ is defined as $\varphi_{\Delta HONO\ and\ \Delta NO_x} \cdot \dfrac{(\gamma_{O_3, Rea} \cdot \omega_{O_3, Rea} - \gamma_{O_3, Ref} \cdot \omega_{O_3, Ref}) \cdot O_{3, amb}}{D}$ represents correction of $\Delta O_3$ due to the uptake loss of $O_3$ on the two chambers. Analogy correction, $\dfrac{(\gamma_{NO_2, Rea} \cdot \omega_{NO_2, Rea} - \gamma_{NO_2, Ref} \cdot \omega_{NO_2, Ref}) \cdot NO_{2, amb}}{D}$, due to uptake loss of NO$_2$ also applies. In our field campaign, correction due to uptake loss of NO$_2$ is much less, relative to the correction associated with O$_3$ uptake loss."

If HONO is generated inside the chamber it will not only release NO (which may not affect the total O$_x$ balance if it just interconvert O$_3$ to NO$_2$) but also OH which most definitely will lead to increased O$_3$ production inside one of the chambers. I don't see how the authors can be sure that HONO production on the chamber walls is not an issue in their system (lines 304).

Response: We have further clarified this point. Please refer to our revision in lines 504-537 "In the 1-week HONO production experiment, uptake loss of HONO during the nighttime were surprisingly spotted, together with production of HONO during the daytime (Fig. 5b). Nighttime loss of HONO on the wall surface ($\Delta$HONO = HONO$_{amb}$ – HONO$_{inchamber}$, $\Delta$HONO = 0.14 ($\pm$0.28) and 0.004 ($\pm$0.30) ppbv for the reaction chamber and reference chamber relative to the ambient), is not suspected. Sklaveniti et al. (2018) reported HONO production rate of up to 9 ppbv h$^{-1}$ with uptake loss of NO$_2$ of 66 ppbv h$^{-1}$ under dark conditions, giving a yield of HONO of 0.14. Even assuming a yield of 0.14 from uptake of NO$_2$, production of HONO would reach 0.42 and 0.28 ppbv h$^{-1}$ in the reaction and reference chamber, *ca.* $\Delta$HONO = −0.14 and −0.093 ppbv. HONO uptake on aerosol particles at night might account for the HONO loss here (Ren et al., 2020). RH in the reaction chamber scatters at approximately 50% ($\pm$13%), which might lead to deliquescence of deposited aerosol particles on the wall surface. As the temperature rose and RH dropped in the early morning, uptake loss of HONO on the wall surface is diminishing. Further decrease in the zenith angle even led to production of HONO in both chambers, resulting in ($\Delta$HONO = −0.11 ($\pm$0.16) and −0.27 ($\pm$0.12) ppbv for the reaction chamber and reference chamber). Daytime source of HONO in zero-OPR control experiment has suggested that this source depended on chamber contamination rather than NO$_2$ concentration. Both heterogeneous uptake of NO$_2$ and unknown source of HONO might account for the daytime HONO production. Daytime $\Delta$HONO herein appeared to be much less than previous reports of 20 ppbv h$^{-1}$ for laboratory conditions (Quartz surface, NO$_2$ = 100 ppbv) (Sklaveniti et al., 2018) and of 11–36 ppbv h$^{-1}$ in the ambient of Houston (Quartz surface, NO$_2$ = 50 ppbv) (Baier et al., 2015). The inert Teflon surface coating and possible cleaner chamber in our Mea-OPR system might account for much less daytime $\Delta$HONO.

While NO$_2$ uptake and HONO production mechanism are not totally accounted for, we carry on our theme discussion on their potential perturbation on O$_3$ photochemistry in the chamber. As calculated in Model S1-S2, in which ambient HONO (HONO$_{amb}$) and HONO in the reaction chamber (HONO$_{Rea}$) is constrained, O$_3$ production overestimation owing to HONO production in the reaction chamber is 5.4% on average during the 1-week HONO production experiment (not shown). The influence of HONO production on Mea-OPR is much weaker than previous reports (Baier et al., 2015; Sklaveniti et al., 2018). More detailed exploring of O$_3$ photochemistry suggests that HONO photolysis comprises 17.6% of the total primary RO$_x$ source budget for the ambient, while comprises 23.0% of the total primary RO$_x$ source budget for the reaction

chamber. The overall primary $RO_x$ source budget is averaged 8.8% higher in the reaction chamber than in the ambient air, which reflects that $\Delta HONO$ does considerably perturb the $RO_x$ source budget and therefore $O_3$ production. In our model, not all $RO_2$ or $HO_2$ results in $O_3$ production (Tan et al., 2018b; Ma et al., 2022), which accounts for lower percent of Mea-OPR perturbation than $RO_x$ budget perturbation by HONO production. In the previous study in Houston, HONO photolysis comprises 29% of the total primary $HO_x$ source budget for the ambient, while comprises 40% of the total primary $HO_x$ source budget in the reaction chamber (Czader et al., 2012). Much less contribution of HONO photolysis to overall production of peroxy radical in Beijing (Lu et al., 2013) and much less $\Delta HONO$ for our Mea-OPR therefore accounts for much weaker perturbation on OPR in the reaction chamber. As calculated in Model S3, in which $HONO_{Rea}$ and $NO_{x, Rea}$ are constrained, $O_3$ production overestimation in the reaction chamber is 9.4% (Fig. 5c)."

Minor Comments
* * *
Equation 5: how is this derived?

Response: Please refer to line 266-282 "Mea-OPR can be calculated in Eq. (5), which is modified from previous scheme in the literature (Cazorla and Brune, 2010; Sadanaga et al., 2017; Sklaveniti et al., 2018). $\Delta NO_2$ and $\Delta O_3$ are the differential $NO_2$ and $O_3$ between the two chambers, respectively. D is the diameter of chambers in m. $O_{3, amb}$ and $NO_{2, amb}$ represent the ambient $O_3$ and $NO_2$ concentration in ppbv, respectively. $\varphi_{trans}$ is the ratio of in-chamber $j(O^1D)$ to ambient $j(O^1D)$ as determined by the UV transmittance of the two Mea-OPR chambers. We assume linear dependence of OPR on $j(O^1D)$ (Tan et al., 2018a), and therefore Mea-OPR underestimation on ambient OPR associated with chamber filter of UV in the reaction chamber and stray light in the reference chamber can be corrected by $\varphi_{trans}$. Positive $\Delta HONO$ in the daytime is found in our field campaign due to $NO_2$ uptake or unknown chamber source of HONO (Rohrer et al., 2005; Akimoto et al., 1987; Ye et al., 2016). As a result, Mea-OPR tends to be overestimated with a correction factor of $\varphi_{\Delta HONO}$ typically higher than unit. Negative $\Delta NO_x$ is observed in our field campaign due to uptake loss of $NO_2$ on chamber wall. This leads to bias in $\Delta O_x$ measurements and also reaction chamber underestimation of ambient $O_3$ production. As a result, Mea-OPR tend to underestimate OPR with a correction factor of $\varphi_{\Delta NO_x}$ typically lower than unit. In addition, chamber formation of HONO is somewhat associated with uptake loss of $NO_2$, therefore $\varphi_{\Delta HONO}$ and $\varphi_{\Delta NO_x}$ are evaluated together. The overall correction factor associated with imperfect chamber mimic of ambient $O_3$ photochemistry due to $\Delta HONO$ and $\Delta NO_x$ is defined as $\varphi_{\Delta HONO \text{ and } \Delta NO_x} \cdot \dfrac{(\gamma_{O_3, Rea} \cdot \omega_{O_3, Rea} - \gamma_{O_3, Ref} \cdot \omega_{O_3, Ref}) \cdot O_{3, amb}}{D}$ represents correction of $\Delta O_3$ due to the uptake loss of $O_3$ on the two chambers. Analogy correction, $\dfrac{(\gamma_{NO_2, Rea} \cdot \omega_{NO_2, Rea} - \gamma_{NO_2, Ref} \cdot \omega_{NO_2, Ref}) \cdot NO_{2, amb}}{D}$, due to uptake loss of $NO_2$ also applies. In our field

campaign, correction due to uptake loss of $NO_2$ is much less, relative to the correction associated with $O_3$ uptake loss."

Figure S1: I would suggest to move this figure to the main text. And also to add a proper description of the instrument in the main text.

Response: Suggestion is accepted!